# Active Slice Discovery in Large Language Models

**Minhui Zhang**[*]
University of Waterloo
anna.zhang@uwaterloo.ca

**Prahar Ijner**[*]
University of Waterloo
p2ijner@uwaterloo.ca

**Yoav Wald**
New York University
yoav.wald@nyu.edu

**Elliot Creager**
University of Waterloo
creager@uwaterloo.ca

## Abstract

Large Language Models (LLMs) often exhibit systematic errors on specific subsets of data, known as *error slices*. For instance, a slice can correspond to a certain demographic, where a model does poorly in identifying toxic comments regarding that demographic. Identifying error slices is crucial to understanding and improving models, but it is also challenging. An appealing approach to reduce the amount of manual annotation required is to actively group errors that are likely to belong to the same slice, while using limited access to an annotator to verify whether the chosen samples share the same pattern of model mistake. In this paper, we formalize this approach as *Active Slice Discovery* and explore it empirically on a problem of discovering human-defined slices in toxicity classification. We examine the efficacy of active slice discovery under different choices of feature representations and active learning algorithms. On several slices, we find that uncertainty-based active learning algorithms are most effective, achieving competitive accuracy using 2-10% of the available slice membership information, while significantly outperforming baselines.

## 1 Introduction

Single errors in machine learning models, and Large Language Models (LLMs) in particular, are often representative of a wider pattern. For example, when asking GPT-5 "Which city is further north? London or Montreal? Answer in one word." we observe the response "Montreal."[2] The correct answer is in fact London, and we may wonder whether the model's responses generally tend to associate locations with cold climates, such as Montreal, to being northern than those that are relatively warmer. More generally, discovering patterns or groups of examples where models underperform, also called error slices, is useful in many aspects, e.g. guiding future data collection and model development.

The problem of discovering coherent groups of examples that a model tends to get wrong is known as *slice discovery* [1, 2, 3]. Most algorithms for this problem work in a completely unsupervised setting: namely, slice discovery algorithms are provided with a set of error cases and are tasked with identifying underlying semantic groups. This is well motivated by real-world applications. Error cases may be reported and collected in a dispersed manner, and grouping them into semantically coherent subsets is laborious. However, the unsupervised setting is challenging and solutions that involve some mechanism of supervision can be attractive. In this paper, we initiate the study of an

---

[*]Equal Contribution
[2]https://chatgpt.com/share/68af39b1-3c54-8008-848d-a69f63abf5eb

39th Conference on Neural Information Processing Systems (NeurIPS 2025) Workshop: Reliable ML from Unreliable Data.

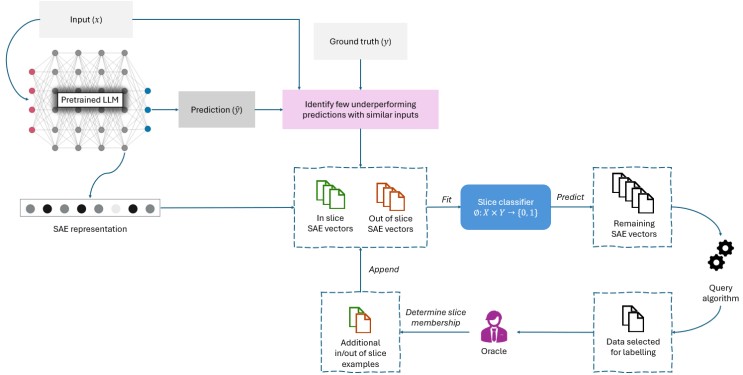

Figure 1: **Active Slice Discovery** Given a dataset containing target labels for each data point, but slice labels for a limited set of data points, we pose the active learning problem of uncovering latent error slices within the data. The active learner has limited access to an oracle (e.g. human labeler) who can confirm whether or not a specific data point belongs to the error slice.

active learning [4] approach to this problem, which we call *active slice discovery*. For example, an active slice discovery algorithm might suggest asking for a one-word answer to the query "If I move from Ulaanbaatar to Paris, should I go north or south?", to which GPT-5 also gives the incorrect answer "South". This $(x, y)$ pair of prompt and response will be moved to a human annotator who will confirm that this is indeed an error and it is semantically similar to the first one. The procedure proceeds iteratively to leverage the enlarged labeled set and improve the characterization of an error slice. Although the slice in this example is relatively straightforward to characterize and discover, the problem can become more complicated as we deal with larger datasets, multiple slices, and more subtle error patterns. We wish to assess the potential of active slice discovery as a practical methodology and form best practices in applying it. To this end, our contributions are:

- We state and initiate the study of the active slice discovery problem, laying the foundation for further work on this attractive approach.

- We implement a flexible active slice discovery pipeline, where combinations of different base models (LLMs), representations, classifiers and active learning strategies can be used. The source code will be made available upon publication to support future research on the problem.

- Our experiments study active slice discovery on the problem of toxicity classification, using the Jigsaw toxicity dataset and Llama 3.1. Comparing several active learning techniques, representations, and classification methods, our results show that uncertainty based active learning methods can reach comparable accuracy to that obtained with the full training dataset using as few as $2\%$ of the labels, and present a significant improvement w.r.t a random subsampling baseline.

## 2   Related Work

**Slice Discovery.** Motivated by known sensitivity to distribution shift [5], slice discovery seeks to group systematic model errors to explain why, for instance, image classifiers overly rely on background features. Prior work explored slice discovery in a variety of use cases, from structured and tabular data [6, 7, 8] to larger computer vision or text models [2, 1, 9, 3], even summarizing inferred slices with textual descriptions [1, 10]. Work with neural network representations obtained from hidden layers of neural networks is a prominent paradigm in slice discovery, and more generally, the interpretability literature.

**Interpretability of LLMs.** In the context of Large Language Models, analyzing internal representations to uncover features is a key goal of mechanistic interpretability [11], which relates to slice discovery, as both tasks involve recovering features and slice discovery can be considered as a specific type of interpretability task. A popular method in this context is to use Sparse Auto-Encoders (SAEs) [12, 13]. Drawing inspiration from the mechanistic interpretability literature, our experi-

ments examine whether representations obtained from SAEs are beneficial for slice discovery.

**Active Learning and Uses in Slice Discovery** In all slice discovery settings mentioned above, the learner is not given annotations of any slices. We study the case where it is possible to elicit a small amount of human judgments on membership of examples in a common slice, a type of auditing that is most closely related to active learning [4]. The vast literature on active learning techniques, such as those based on uncertainty of the model $f(x)$; diversity of the labeled set [14]; coresets [15], and others, deals with eliciting labels $y$ that are most useful in training an accurate prediction model. In the context of slice-discovery, perhaps the closest work to ours, is [16] that uses slice discovery to detect error-prone examples that guide active learning to train a more accurate classifier. While their framework uses common slice discovery methods without slice annotations and elicits labels $y$, the active slice discovery we explore here elicits slice identities $\mathbf{s}$. That is, we seek more accurate slice discovery with the fewest annotations possible, while they seek accurate classification with fewest labels possible as in common active learning settings.

# 3  Problem Definition

Formally, we consider a joint distribution $P(X, Y, \mathbf{S})$ over the space $\mathcal{X} \times \mathcal{Y} \times \{0, 1\}^k$ for some integer $k$, where all data is drawn $i.i.d$ from $P$. Here $X$ is the input text, $Y$ is a label (e.g. toxic/non-toxic) and $\mathbf{S} = \{S^{(i)}\}_{i=1}^k$ is a vector of slice memberships for $k$ slices of interest.

- **Input:** a trained classifier $f_\theta : \mathcal{X} \to \mathcal{Y}$, small annotated dataset $\mathcal{D}_s = \{(x_i, y_i, \mathbf{s}_i)\}_{i=1}^{n_s}$, and labeled dataset $\mathcal{D} = \{x_i, y_i\}_{i=1}^n$ drawn i.i.d from $P(X, Y, \mathbf{S})$ and the marginal $P(X, Y)$ respectively. We also have a budget $K$ of active slice queries.
- **Output:** A slice membership function $\phi : \mathcal{X} \times \mathcal{Y} \to \{0, 1\}^k$.

An active slice discovery method has a query strategy $A : \{\mathcal{X} \times \mathcal{Y} \times \{0, 1\}^k\}^{n_s} \times \{\mathcal{X} \times \mathcal{Y}\}^n \to [n]$ that takes the currently annotated dataset $\mathcal{D}_s$, and labelled dataset $\mathcal{D}$, and chooses a specific unlabeled example,or "query", $(x, y) \in \mathcal{D}$ to be annotated. The accuracy of $\phi_j$, the component that detects the $j$-th slice, is $\mathbb{E}_{x,y,\mathbf{s}}[\mathbf{1}[\phi_j(x, y) = s_j]]$.

# 4  Experiments

We evaluate the proposed active slice discovery approach on the Jigsaw Toxicity dataset, [17], and consider the use of two possible internal state representations: (1) raw layer embeddings from the penultimate layer of Lamma-3.1-8B, and (2) sparse activations obtained from the Llama Scope sparse autoencoder trained on the final layer of Llama-3.1-8B. For slice classification, we compare a feed-forward multi-layer perceptron (MLP) against a linear support vector machine (SVM). We leverage the Small-Text library [18] to explore established Active Learning query strategies.

## 4.1  Sample Efficiency of Active Learning

We start the evaluation by examining how sample efficiency depends on the type of slice. Holding the representation (SAE), OVR classifier (SVM), and query strategy (Least Confidence) fixed, we vary the slice definition (e.g., disagree, likes). The results from Figures 2a and 2b indicate that identity based slices like *female*, *christian* can be detected with high accuracy using only a few hundred annotations, whereas some reaction based slices like *disagree* and *sad* fail to significantly improve with under 1000 labeled samples. This indicates that slices with similar lexical cues can be easier to identify compared to heterogeneous and sentiment based slices. However, we note that the detection rate for the *disagree* slice improves significantly from 0.8 with layer embeddings to 0.83 with SAE representations.

## 4.2  Effect of Query Strategy

Next, we compare different query strategies for active learning by fixing the slice to the *disagree* slice. The query strategies studied included uncertainty-based strategies (*Least Confidence*, *Prediction Entropy*, *Breaking Ties*), diversity-based strategies (*Embedding K-Means*, *Discriminative Active Learning*, *Lightweight Coreset*), and a baseline *Random Sampling* strategy. The queries are tested

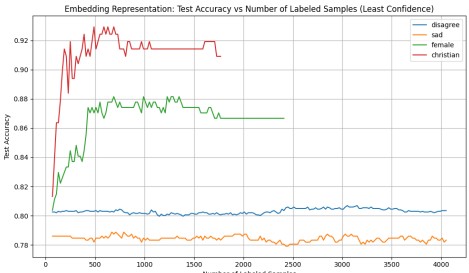

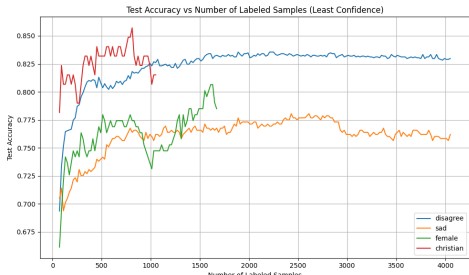

(a) Active learning with SVM (Least Confidence) using raw LLM embeddings.

(b) Active learning with SVM (Least Confidence) using SAE representations.

Figure 2: **Active learning with SVM (Least Confidence) on multiple slices.** Test accuracy vs. number of labeled examples is shown for two setups: (a) raw LLM embeddings and (b) SAE representations. Note that each of the four slides is a different size, leading of a different max number of labeled samples.

across the raw layer embeddings and SAE based representations as represented in Figure 3a, 3b. The results show that uncertainty based query strategies yield higher accuracy with fewer labels across both embeddings and SAE based inputs. This finding aligns with prior work in text classification active learning [19]. The use of SAE features further improves the stability of the active learning training process, yielding smoother training curves and being less sensitive to the uncertainty query strategy.

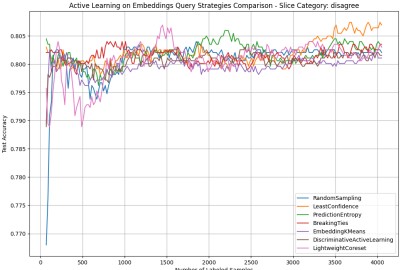

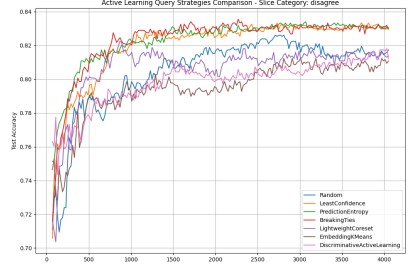

(a) Active slice discovery from LLM embeddings

(b) Active slice discovery from SAE features

Figure 3: **Query strategy comparison for the "disagree" slice.** Test accuracy vs. number of labeled examples is shown for various query strategies acting on (a) raw LLM embeddings; and (b) SAE representations. Confidence-based query strategies (Least Confidence, Prediction Entropy, Breaking Ties) consistently yield better performance.

| Setup | Slice Classifier | Best Accuracy | Labeled Examples |
|---|---|---|---|
| Setup 1 (Embedding) | Neural Network (AL) | **85.8**% | 250 (out of 12,504) |
| Setup 1 (Embedding) | SVM (AL, LC query) | 81.0% | 3,500 |
| Setup 2 (SAE) | Neural Network (AL) | 82.2% | 1,460 (out of 12,416) |
| Setup 2 (SAE) | SVM (AL, LC query) | 83.0% | 1,000 |

Table 1: **Active slice discovery performance on the "disagree" slice**. We report the highest test accuracy achieved by each method and the number of labeled training examples required. (LC = Least Confidence query strategy.)

## 4.3 Combined Results

Table 1 summarized the strongest results observed across all configurations. Overall, the MLP achieves the highest observed accuracy (*85.8%* detection accuracy using raw layer embeddings with active learning. However, this approach requires careful hyperparameter tuning. Alternatively, using an SVM with SAE input features is simpler to train, requiring little to no hyperparameter tuning, and achieves a competitive accuracy (*83.0%* while using more labelled examples. These findings highlight two practical observations: first, active learning can reduce labeling requirements by up to 98% relative to full supervision; second, high-quality representations such as SAEs allow even simple models to remain competitive.

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
