# OpenReview forum: "Active Slice Discovery in Large Language Models"
_NeurIPS.cc/2025/Workshop/Reliable_ML — NeurIPS 2025 - Reliable ML Workshop_

### Official Review · Reviewer_oRXS · 2025-09-18
**The paper is novel, clear and relevant to reliability with imperfect data. Clear accept.**

**Rating:** 8
**Confidence:** 3

**Review:**

The authors of the paper state and initiate the study of active slice discovery in LLMs, that is, actively grouping errors that are likely to belong to the same error slice, while using limited access to an annotator to verify whether the chosen samples share the same pattern of model mistake. The purpose of active slice discovery is to reduce the amount of manual annotation of examples needed. Comparing several combinations of different base models (LLMs),
representations (raw layer embeddings, SAE), classification methods (MLP, SVM) and learning strategies, the authors conclude that uncertainty based active learning methods can reach comparable accuracy to that obtained with the full training dataset using as few as two percent of the labels.

The paper appears to be novel and is relevant to reliability with imperfect data and the arguments are presented with clarity. No weaknesses observed.

---

### Official Review · Reviewer_CzeG · 2025-09-20
**Clear argument for active error slice discovery in LLMs**

**Rating:** 6
**Confidence:** 3

**Review:**

This paper discusses active error slice discovery to find systematic problems in large language models. Their experiments centered around a toxicity dataset where they looked at accuracy versus labeled samples. They compared and contrasted different querying and model options with active learning, and they found that SVM with SAE inputs remains competitive against neural networks even with little hyper parameter tuning.

Pros:
- Paper is well-written and easy to understand. The examples provided are clear and further strengthen the writing.
- These findings draw practical insights from experimental results.
- Experiments make sense for their problem and are easy to follow.
- Proposed approach is interesting and tackles a clear problem of discovering error slides as LLMs outputs become increasingly complex.

Cons:
- Labels on graphs are very small.
- I wish Table 1 also included some comparison to another paper doing error detection. Having some sort of confidence interval around accuracy would have also been helpful and may have also strengthened your argument about smaller models remaining competitive.
- I think Figure 3 could have been shown differently. It takes up a lot of space, and there are too many colors to draw meaningful comparisons. I think plotting the difference between the two setups would have been useful here. The extra space could have been added to make a more clear limitations/conclusions section.